# The Management between Comorbidities and Pain Level with Physical Activity in Individuals with Hip Osteoarthritis with Surgical Indication: A Cross-Sectional Study

**DOI:** 10.3390/medicina57090890

**Published:** 2021-08-27

**Authors:** Michael Silveira Santiago, Felipe J. Aidar, Talita Leite dos Santos Moraes, Jader Pereira de Farias Neto, Mário Costa Vieira Filho, Diego Protásio de Vasconcelos, Victor Siqueira Leite, Felipe Meireles Doria, Erick Sobral Porto, Reuthemann Esequias Teixeira Tenório Albuquerque Madruga, David Edson Ramos de Azevedo, Adonai Pinheiro Barreto, Marcel Vieira Gomes, Paulo Francisco de Almeida-Neto, Breno Guilherme de Araújo Tinôco Cabral, Walderi Monteiro da Silva Júnior

**Affiliations:** 1Program of Physical Education, University Hospital of Sergipe (UFS), São Cristovão 49100-000, SE, Brazil; michael.santiago@ebserh.gov.br (M.S.S.); talita.leite.fisioterapia@gmail.com (T.L.d.S.M.); diego.protasio@ebserh.gov.br (D.P.d.V.); deratrainer@gmail.com (D.E.R.d.A.); walderim@yahoo.com.br (W.M.d.S.J.); 2Department of Physical Therapy, University Hospital of Sergipe (UFS), São Cristovão 49060-025, SE, Brazil; jaderneto@academico.ufs.br; 3Group of Studies and Research of Performance, Sport, Health and Paralympic Sports (GEPEPS), Federal University of Sergipe (UFS), São Cristovão 49100-000, SE, Brazil; victor_s_leite@hotmail.com (V.S.L.); felipemdoria@gmail.com (F.M.D.); 4Department of Physical Education, Federal University of Sergipe (UFS), São Cristovão 49100-000, SE, Brazil; 5Program of Physiological Science, Federal University of Sergipe (UFS), São Cristovão 49100-000, SE, Brazil; 6Department of Medicine, Federal University of Sergipe (UFS), São Cristovão 49100-000, SE, Brazil; mcvf99@academico.ufs.br (M.C.V.F.); fattam@uol.com.br (R.E.T.T.A.M.); 7Department of Medicine, Tiradentes University (UNIT), Aracaju 49032-490, SE, Brazil; ekporto@hotmail.com; 8Department of Orthopaedics, University Hospital, Federal University of Sergipe (UFS), Aracaju 49060-025, SE, Brazil; adonai.barreto@ebserh.gov.br; 9Program of Traumatology and Orthopaedics, Federal University of Sergipe (UFS), São Cristovão 49100-000, SE, Brazil; vieiramarcelgomes@gmail.com; 10Health Sciences Center, Federal University of Rio Grande do Norte, CCS-UFRN, Natal 59012-570, RN, Brazil; paulo.neto.095@ufrn.edu.br (P.F.d.A.-N.); brenotcabral@reitoria.ufrn.br (B.G.d.A.T.C.); 11Department of Physical Education, Federal University of Rio Grande do Norte, DEF-UFRN, Natal 59078-970, RN, Brazil

**Keywords:** hip osteoarthritis, pain, physical activity

## Abstract

*Background and Objectives*: The degenerative pathology of the hip joint appears in young age groups, related to fem-oroacetabular impingement, and in advanced age, due to other inflammatory causes, with greater potential for severity in the presence of comorbidities. *Objectives*: To evaluate the participation of the main causes of osteoarthritis in relation to physical activities, s Body Mass Index (BMI) and television time (TV). *Materials and Methods*: 54 patients with surgical indication treated at an orthopedic referral university hospital were stratified into groups (Impact: I, Osteonecrosis/rheumatic: II, Infectious/traumatic: III), and the influence of comorbidities on physical activity performance, relative to BMI and TV time. *Results*: It was observed that the impact group was the most frequent (51.8%), with 79.6% under the age of 60 years. This group followed the general mean (*p* < 0.05), using the variables of comorbidity and the level of physical activity. Pain intensity, TV time, BMI showed no correlation with physical activity. *Conclusion*: Morphostructural changes (group I) represented the most frequent etiological group, and severe pain was common in almost the entire sample. Unlike BMI, comorbidity showed a significant relationship with the level of physical activity.

## 1. Introduction

Osteoarthritis (OA) is a chronic musculoskeletal inflammatory disease that compromises the joints and generates progressive chondral damage, with the proliferation of subchondral bone and reactional synovitis [1]. It affects more than 30% of the general population over the age of 30 and more than 65% of those over the age of 65 [2]. This pathology does not have a sustained therapeutic resolution given the poor response to clinical measures, allowing joint reconstruction surgeries to be increasingly performed [3]. The painful symptoms and the gradual loss of segment function are individual and variable depending on the severity of the joint pathology, by the presence of comorbidities and the perception by the patient or the examining physician [4,5,6]. The high prevalence of OA and the negative impact on functionality and life quality increase the importance of identifying the factors that influence the prevention and control of symptoms [7].

The hip joint is one of the joints most affected by OA, with a lifetime risk (until 85 years) of 25% [8]. The appearance of this joint disease in younger age groups increased due to the early diagnosis of morphostuctural malformations [9]. The prevalence is higher in young men and in women aged over 65. As it is a weight-bearing joint, there is a rapid progression of joint degeneration, with a greater implication in mobility. About 80% of patients with this clinical condition have some movement deficit [10]. Of these, 25% are unable to perform daily life activities [11,12,13]. Because of the progressive loss of hip functionality, it commonly restricts mobility in daily activities, which makes it difficult to start or maintain exercises. With the highest severity of OA (degrees ≥ 2 by the Kellgren–Lawrence radiographic classification), joint degeneration becomes more limiting [14]. The functionality of this segment determines a variable degree of physical activity and its practice in all age groups, and aims to modify the lifestyle and prevent chronic diseases [15]. These pathologies make it difficult to participate in physical activities due to the impairment of locomotion, lower limb vasculopathy, and heart disease [16]. The broader components of this activity are related to working time, transportation, domestic life, and leisure situations, which consist of exercises and recreational sport from low to high impact [17].

With the evolution of these various types of activities, there have been significant changes in their performances, from those developed for entertainment to the regular practice of exercises, with different goals depending on the individual lifestyle [18]. There is evidence that exercises of light to moderate intensity do not influence the worsening of joint pain and do not accelerate the pathological process of OA [19,20]. Changes in the type of load on the joint or different types of physical activity can generate different responses, depending on the intensity of the exercise, duration, and age at the beginning of the practice of the activity [21]. The Johnsen [22] study showed an association between high-impact physical activity and increased risk of OA. On the other hand, physical inactivity (immobilization, sedentarism) also results in a catabolic response, decreasing the thickness of the cartilage [23]. The lack of exercise (even when of low impact) influences the appearance or the increase in the severity of common chronic diseases, such as diabetes mellitus, cancer, cardiovascular disease, and osteoporosis [24,25]. The causes of hip osteoarthritis range from morphological changes (anatomical, structural), rheumatic conditions, the femoral head′s osteonecrosis, and even infectious and traumatic events.

Different to osteoarthritis located in other joints, the hip is a common site of osteochondral damage without the concomitant presence of damage to other segments [26]. There are specific local factors that can initiate the process of joint degeneration, so that hematological changes, rheumatic processes, previous infectious conditions, and structural anatomical deformities trigger bone or chondral injury [27]. There are changes in hip, femoral, or acetabular geometry which predispose to mechanical conflict (impact) and which can trigger the onset of osteoarthritis [28]. Structural and morphological deformities in the hip joint may be caused by acetabular dysplasia, the loss of sphericity of the femoral head (in the anterolateral portion of the femoral head–neck transition), may be secondary to proximal femoral epiphysiolysis or post-traumatic sequelae [29]. In advanced stages of hip degeneration, it is not possible to perform the radiographic views that allow one to measure the α angle (between the neck axis and the bone point of the femoral head–neck junction) and LCE angle (lateral center-edge), which allow one to diagnose femoroacetabular impingement and dysplasia [30,31]. In advanced osteoarthritis, there is a deformity (more evident) in the femoral head, a great reduction in the joint space and the formation of osteophytes in the femur and acetabulum [9,32].

There is conflicting evidence that the worsening of the painful hip condition is related to the predictors: gender, age, body mass index, the duration of symptoms, and the Kellgren–Lawrence radiographic classification [14]. According to the study [33], age, joint space width, femoral head migration, femoral osteophytes, bone sclerosis, and hip pain are associated with progression of hip osteoarthritis. Unlike gender and body weight, it is not related to osteoarthritis in this segment. Associations between body mass index and hip osteoarthritis risk do not vary by sex, study design, or osteoarthritis definition [34]. When considering that pain and decreased functionality of the hip are the main symptoms in osteoarthritis, scientific evidence has shown that this clinical worsening has a strong relationship with the presence of comobidities, poor general condition, and low vitality. The presence of comorbidities represents a variable that can influence the participation or practice of physical activity; however, it shows inconsistent data [35]. The application of this variable in a group with morphostructural changes is not clarified in the scientific literature [36,37]. Therefore, the primary objective of this study is to assess the incidence of pathologies with morphostructural (from impact) changes, relating to comorbidities and the practice of physical activity. Additionally, a secondary objective is to perform a descriptive analysis of pain intensity, body mass index, and television time (sedentary behavior) according to each group of hip OA etiology.

## 2. Materials and Methods

### 2.1. Design

This was a cross-sectional study carried out at the orthopedics outpatient clinic of the hip joint reconstruction group at the University Hospital of the Federal University of Sergipe (UFS) with 54 subjects who presented hip osteoarthritis, clinically confirmed by the presence of intra-articular pain in the groin area, lateral region of the hip with the possibility to extend from the thigh to the knee, and reduced mobility at the expense of flexion, adduction and internal rotation. Due to the difficulty in surgically solving hip pathology in a public hospital, a cross-sectional study was chosen to better perform the descriptive analysis of a specific profile of patients.

Radiographically, the Kellgren–Lawrence classification [14] was used, with degrees that vary according to joint narrowing and osteophytes′ presence, with degrees > 2 showing a reduction in joint space greater than 2 mm. as the following were adopted as inclusion criteria: the radiographic classification (mentioned above) and adults over 20 years old. The exclusion criteria were met by individuals who did not undergo clinical follow up through the application of questionnaires and clinical examination directed at the hip, motor limitation that prevented the performance of any of the evaluations, or cognitive impairment due to having Alzheimer′s disease, cerebral palsy or any neurological disease that impairs verbalization. Hips with advanced osteoarthritis were evaluated, and the diagnosis of the pathology was previously discriminated by clinical and radiographic follow up. Osteonecrosis of the femoral head due to idiopathic or hematological causes, femoroacetabular impingement, sequelae of articular infectious conditions, and proximal femoral epiphysiolysis, dysplasia and acetabular protrusion were the primary diagnoses. Patients with these pathologies were evaluated according to individual variables [38].

### 2.2. Sample

A convenience sample was used with 54 patients seen at a hip clinic, after meeting the inclusion criteria. All the patients were eligible and were on the waiting list for arthroplastic surgical intervention and remained under orthopedic clinical treatment under drug support and rehabilitation (when they tolerated it) to control pain and attempt to improve functionality. We included this group of patients because, as they were patients from the public health system, with poor clinical follow up in view of the orthopedic appointments, they sought out the university reference service in more advanced stages of osteoarthritis.

Sampling power was calculated a priori using the open-source software G * Power^®^ (Version 3.0; Berlin, Germany), choosing a “T-family statistic (Correlations)”, considering a pattern α < 0.05, β = 0.80 and the effect size of 0.39 found for the relationship between the level of physical activity and the degree of OA in patients of both sexes in the study by Morcos et al. [39]. Thus, it was possible to estimate a sampling power of 0.80 (t (3.4): 1.69) for a minimum sample of thirty-six subjects (*N* = 36), suggesting that the sample size of the present study had statistical strength from the research approach. For comparisons, we used the mean data from the study by Rosemann et al. [40]; those authors compared the different levels of physical activity (vigorous, moderate, and mild) in patients with OA. In this sense, we used the open-source software G * Power^®,^ choosing a “F family statistic (ANOVA-one way comparisons)”, considering a pattern α <0.05, β = 0.80; thus, through the function “ Determine” based on the reported averages, the software indicated an effect of 4.7, which we adopted for the analyses. Thus, a minimum sample number of 6 subjects per group was indicated for a sampling power of 0.99 (F(3.0): 9.55).

Patients participated in the study on a voluntary basis and signed a free and informed consent form, according to resolution 466/2012 of the National Research Ethics Commission—CONEP, of the National Health Council, in accordance with the ethical principles expressed in the Helsinki Declaration (1964, reformulated in 1975, 1983, 1989, 1996, 2000, 2008 and 2013). The project was referred to the Research Ethics Committee of the Federal University of Sergipe with CAAE: 16843619.4.0000.5546 and approved with the following opinion: 3,623,993 in 2019.

### 2.3. Instruments

#### 2.3.1. Anthropometric Data

BMI (body mass index) was obtained by dividing the weight in kg by the height in meters. For the clinical examination by the researching physician, the patients remained in a supine position to assess the deficit in joint amplitude in flexion, extension, internal and external rotation, abduction, and adduction. In the radiographic evaluation, the anteroposterior views and ducroquet profile (flexion of 90 degrees and abduction of 45 degrees) were used to better visualize the joint space or morphological (structural) changes, femoral head necrosis, and concentric or eccentric joint reduction depending on the location of the femoral head involvement [5,41].

#### 2.3.2. International Physical Activity Questionnaire (IPAQ), Long Form

Four domains (domestic and work occupation, leisure, transport) were used as questions. We recorded the number of minutes that the patients remained standing, walking or any other activity per day and multiplied these by 7 days. Those who carried out more than 150 min of activity per week were considered active, while those who carried out less than 150 min were considered inactive [42,43].

#### 2.3.3. Visual Analogic Scale (VAS)

The visual analogic scale was used to assess the level of pain in individuals. It consists of a variation of 11 numbers, between 0 and 10, where 0 is considered no pain, and 10 the maximum unbearable pain. This evaluation was carried out as soon as the patient was admitted to the study [44].

### 2.4. Proceadure—Registration Form

Television time (hours spent sitting or lying watching television) was also recorded in the registration form [45].

As the university hospital is a reference for chronic degenerative pathologies, we updated the data with the registration of comorbidities: rheumatic, cardiac, traumatic, and hematological. The type of physical activity developed was classified as: no activity, domestic, leisure (walking, cycling, swimming, stretching), and labor occupation. Due to the need to evaluate the cause of hip osteoarthritis in each subject, we grouped the etiologies into three groups, according to the proximity of the form of presentation. The first group represented those with morphological changes (femoroacetabular impingement, sequelae of epiphysiolysis, and developmental dysplasia of the hip). The second group was formed by rheumatic pathology, osteonecrosis of the femoral head induced by corticosteroid therapy or by hematological disease, and acetabular protrusion. The last group included traumatic and infectious causes. For each group, the same data were collected so that we could better understand the relationship of these etiologies with comorbidities or with the level of physical activity. In these groups, we did not measure the α and CE angles, as they presented advanced osteoarthritis [30,31]. Comorbidities were recorded as soon as they were included in the study, the most frequent being essential arterial hypertension, diabetes mellitus, systemic lupus erythematosus, asthma, and cirrhosis. All cases of osteoarthritis were in advanced degrees with great loss of joint function by clinical examination and typical radiographic signs of arthrosis.

### 2.5. Statistical Analysis

Descriptive statistics of the characterization of the study were made, using the measures of central tendency, average (X) ± standard deviation (SD) and 95% confidence interval (95% CI) with the dependent variables (level of physical activity) and the determinants (BMI, age, sex, time of most intense pain, comorbidities, television time). To check the normality of the variables, the Kolmogorov–Smirnov (KS) test was used, given the sample size (54 subjects). The data for all variables analyzed were homogeneous and normally distributed. To assess the difference between the type of physical activity and the presence of comorbidities, the ANOVA test (one way) was performed, with Bonferroni′s post hoc being applied to the general group of 54 subjects and to groups I, II, and III. Binary logistic regression was performed, using categorical variables, age (dichotomy at 40 years old), and the level of physical activity. The statistical treatment was performed using the computerized package Statistical Package for the Social Science (SPSS), version 22.0. The level of significance was set at *p* < 0.05. To check the effect size, (Eta partial squared: η^2^p), values of low effect (≤0.05), medium effect (0.05 to 0.25), high effect (0.25 to 0.50), and very high effect (>0.50) were adopted for ANOVA [46].

## 3. Results

Group I (morphostructural changes) represented 51.8% for the etiology of hip OA, followed by the group with rheumatic pathology, osteonecrosis, and acetabular protrusion, which contributed 35.2%. Meanwhile, traumatic and infectious causes represented 13%. A total of 3.7% of the sample with osteoarthritis had an infectious cause (septic arthritis).

Group I represented patients with advanced radiological alterations compatible with femoroacetabular impingement, due to the deformity in the femoral head–neck and the supero-lateral arthrosis of the hip. Comorbidities were broken down according to the disease reported in the initial registration, meaning most of them were essential arterial hypertension and diabetes mellitus.

Patients with hip osteoarthritis were characterized and are shown in Table 1. The prevalent age group was below 60 years old (79.6%), with no statistical significance between the levels of physical activity in the groups studied.

Of the adopted convenience sample, 55.6% were male and 44.4%, female. According to the IPAQ (long form), no activity (inactive) was practiced by 35.0% of the participants, while the active ones were divided between those who performed leisure activities (33.5%) and household chores (31.5%). The intensity of pain was part of the clinical evaluation using the visual analogic scale and the high-intensity pain represented 51.9% of the complaints presented by the patients, with the night period being the most frequent (83.3%). We did not differentiate the stages of osteoarthritis, since the entire sample group had advanced degrees. Among the cases of osteonecrosis of the femoral head, all already presented marginal head collapse with reduced joint space.

In the one-way ANOVA, with the dependent variable being the level of physical activity and the group variable, the presence of comorbidities, we observed that group I was the only one that followed the general average, maintaining a *p* < 0.05. According to Table 2, the data showed the significance of each group, and in groups II and III, comorbidity did not influence the level of physical activity.

By VAS, most incidences regarding the intensity of pain occurred at night (83.3%). However, these data did not show any correlation with the level of physical activity.

Table 3 shows the comparisons between the three groups concerning BMI, television time and pain through VAS.

Figure 1 shows the comparisons between pain and physical activities in the groups.

There were differences in pain intensity in relation to the impact group (6.54 ± 4.85, 95% CI 5.21–7.86) and in relation to the IT group (9.71 ± 5.06, 95% CI 5.04–14.39) (*p* = 0.020, F 1223, η^2^p = 0.169) (average effect). Muscle strength was reduced (grades 0–2) in all analyzed patients and functionality was very limited, especially in hip flexion, adduction, and internal rotation of the hip.

There were no differences between the other groups in pain. There was no difference in physical activity between the groups (*p* = 0.495, F = 0.944, η^2^p = 0.136, average effect).

We performed a binary logistic regression with the variables age (dichotomized at 40 years) and the level of physical activity, and it was observed that adults under 40 years of age with hip osteoarthritis (regardless of the cause), were more likely to have an active behavior by the IPAQ.

## 4. Discussion

In our study, we found a higher prevalence of individuals with hip osteoarthritis (candidates for arthroplastic surgery) with severe pain in almost all of the sample, with only 1.9% presenting mild pain. When several activities were divided by the IPAQ, it was observed that the domestic occupation contributed to a higher level of physical activity, classifying the subject as active. These data showed similarity with a study that identified the higher prevalence of young people with morphostructural changes developing low-impact activities such as occupational, domestic, or recreational activities [47]. In this sense, younger individuals are expected to have less functional and strength deficits than those of older age, which justifies a higher level of activity in our sample [12].

Unlike other studies that discriminate the level of physical activity by clinical performance (WOMAC), we opted to use the IPAQ (long form) to better stratify the level of activity according to the domains (domestic, occupational, leisure, transport). With this change in the clinical approach, a higher level of activity was identified in individuals with hip osteoarthritis, mainly in the domestic occupation domain [48,49].

High levels of pain represent one of the main clinical findings in the surgical indication in the hip osteoarthritis group and regardless of the cause (groups of our study), influences the medical therapeutic decision. This clinical component is one of the instruments used to assess the surgical need [50,51].

The most prevalent age group found in the research subjects (<60 years) did not show any similarity with the study by Murphy [2]. However, in this group of patients, the participation of the femoroacetabular impingement was an important etiological factor [30], being that this cause represented 44% of causes in our study. In agreement with other articles, there is a higher incidence of the diagnosis of femoroacetabular impingement in a younger age group, whether with anatomical (CAME), acetabular (PINCER) or mixed alterations [47]. In the group with structural hip deformities, hip dysplasia and femoroacetabular impingement (being primary or secondary to the consolidation of proximal femoral epiphysiolysis) predispose one to degenerative joint diseases [52].

When analyzing Table 2, we observed that the television time in the groups proved to be less than 7 h (cutoff point of the television time for sedentary behavior), in agreement with the study by Ku [53] in most of the sample. However, this level of activity did not show statistical significance in the groups.

In Figure 1, we observed that the femoroacetabular pathology has a lower level of pain than in the group of infectious and traumatic pathologies, with statistical significance. In this sense, the study by Kemp [31], shows that hip pain symptoms with this pathology profile are not an impeding factor for physical activity, which is confirmed by the IPAQ of our study among the subjects in group I.

The data from the binary logistic regression in our study were similar to those found by Bennel [54], when evaluating the behavior of this joint disease in athletes under 40 years of age. The one-way ANOVA statistical analysis adopted between the three groups with the variable’s comorbidities and level of physical activity showed a significant relationship (*p* < 0.05) only in the group with morphostructural changes (from impact). This was the first article that presented a comparative analysis with several pathologies and showed that these individuals have a greater tendency to practice physical activity, such as walking, swimming, isometric stretching exercises, and cycling. The study by Veenholf [55] presented similar conclusions to our study, regarding the lack of evidence of individual factors, such as age and BMI with a low level of physical activity.

At the end of the study, we observed some limitations. First, the sample size was small, especially in groups II (rheumatic disease, osteonecrosis, acetabular protrusion) and III (trauma–infectious causes), which made it difficult to perform logistic regression statistics and ANOVA. Thus, the outcome may be underestimated or highly valued for some analyses.

Second, because it is a cross-sectional study, physical activity was recorded in a single moment without an objective instrument for measuring steps or displacements, and there might have been oscillations in activity throughout the day that were not registered. Third, all individuals had a high degree of osteoarthritis, an arthroplastic surgical indication, and a prior recommendation to avoid high-impact activities, which already made it difficult to adhere to or to practice any physical activity, except domestic ones. Authors should discuss the results and how they can be interpreted from the perspective of previous studies and of the working hypotheses. The findings and their implications should be discussed in the broadest context possible. Future research directions may also be highlighted.

## 5. Conclusions

With the final analysis of the data, we observed that the main cause of hip osteoarthritis in individuals who are candidates for arthroplastic surgery is femoroacetabular impingement, associated with the group of morphostructural changes in the head or acetabulum. Since pain represents the most relevant data in the entire sample, the presence of comorbidities showed a significant relationship with the level of activity in this group.

Anthropometric indicators did not influence the level of physical activity in the scenarios, both with BMI, as with age, and television time. In view of the final data of the study, we observed that in advanced hip osteoarthritis, the presence of pain and comorbidities influence the practice of physical activity, so that other therapeutic measures should be offered to improve functionality.

## Figures and Tables

**Figure 1 medicina-57-00890-f001:**
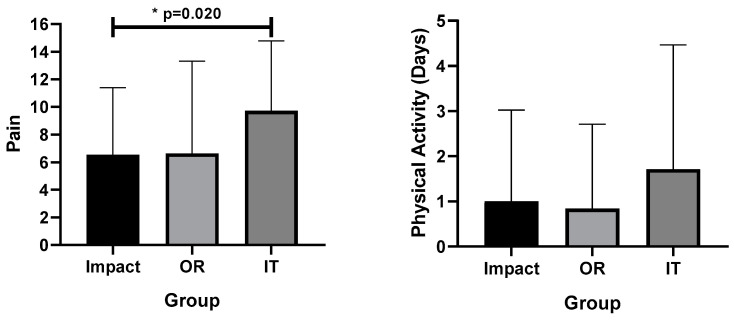
(**A**) Pain and (**B**) physical activity (days) in different pathology etiology types. Legends: OR: osteonecrosis + rheumatic, IT: infectious + traumatic.

**Table 1 medicina-57-00890-t001:** Characterization of the 54 subjects.

	General	Group I	Group II	Group III
Age (Years)	49.90 ± 14.68	49.81 ± 16.22	46.89 ± 12.48	55.14 ± 16.72
BMI	26.79 ± 4.97	26.99 ± 4.83	26.69 ± 5.28	27.19 ± 5.33
osteoarthritis etiology	100%	51.90%	35.20%	13.00%
IPAQ				
Inactive	22.20%	32.10%	10.50%	14.30%
Active	77.80%	67.90%	89.50%	85.70%
Comorbidities				
Without	37.00%	28.60%	47.40%	50.00%
Present	63.00%	71.40%	52.60%	42.90%

Legend: BMI: body mass index; IPAQ: International Physical Activity Questionnaire.

**Table 2 medicina-57-00890-t002:** Level of physical activity X comorbidities.

	Df (Inter Groups)	Df (Intra Groups)	Z	Sig
GENERAL	1	52	10.5	0.02
GROUP I	1	26	6.08	0.02
GROUP II	1	17	2.01	0.17
GROUP III	1	5	0.71	0.43

**Table 3 medicina-57-00890-t003:** Body mass index (BMI), television time (Hours), visual analogic scale (VAS), (average ± SD) in different types of pathology etiology.

	BMI X ± SD (CI 95%)	TV (Hours) X ± SD (CI 95%)	VAS X ± SD (CI 95%)
Impact	28.26 ± 4.96 (23.67–32.85)	5.43 ± 2.30 (3.30–7.55)	6.86 ± 1.35 (5.61–8.10)
Osteonecrosis + Rheumatics	25.31 ± 4.52 (21.13–29.49)	4.29 ± 2.14 (2.31–6.26)	8.14 ± 1.68 (6.59–9.69)
Infectious + Traumatic	27.19 ± 5.33 (22.26–32.12)	3.71 ± 1.98 (1.89–5.54)	6.14 ± 2.19 (4.11–8.17)
*p*	0.838	0.824	0.259
η^2^p	0.072 #	0.169 #	0.277 ##

*p* < 0.05 (ANOVA two-way, and Bonferroni post hoc). η^2^p: # medium effect (0.05 to 0.25), ## high effect (0.25 to 0.50). Legend: BMI: body mass index; TV: television; VAS: visual analogic scale.

## Data Availability

The data that support this study can be obtained from the address: www.ufs.br/Department of Physical therapy.

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
