# Peer review of "The Management between Comorbidities and Pain Level with Physical Activity in Individuals with Hip Osteoarthritis with Surgical Indication: A Cross-Sectional Study"

_medicina, 2021, doi:10.3390/medicina57090890_

Round 1

Reviewer 1 Report

dysplasia and protrusion. Acetabular were the primary diagnoses and we evaluated patients with these pathologies according to individual variables [29]. - review this sentence!

There were  02 patients with osteoarthritis of an infectious cause (septic arthritis). - review 02

Author Response

Thank you for your considerations. The adjustments are in accordance with the attached document

Reviewer 2 Report

Comments

- Add to the title so that you can see that this study is a cross-sectional study.

- Add a reference. “Different of osteoarthritis located in other joints, the hip is a common site of osteochondral damage without the concomitant presence of damage to other segments.”

- It is necessary to clearly state the references to some of the sentence in the introduction. Review the whole thing.

- The author described the following in the introduction. “There is conflicting evidence that the worsening of the painful hip condition is related to the predictors: gender, age, body mass index, duration of symptoms and the Kellgren-Lawrence radiographic classification [10].”, I hope to describe this in more detail.

- The author needs to describe the introduction based on previous research results. The present introduction feels like reading a book. It is hoped that unnecessary sentence will be deleted and that only important sentence will be described based on previous research results.

- You'll need to describe the number in a few headings (For example, 2.3.1. Anthropometric Data, 2.3.2. International Physcal Activity Questionnaire (IPAQ), Long Form, 2.3.3. Visual Analogic Scale).

- What is the clinical significance of this study's results to potential readers?, Describe clinical significance in the conclusion section.

Author Response

(The authors gave the same response as above.)
